# Association of Hippocampal Subfield Volumes with Amyloid-Beta Deposition in Alzheimer’s Disease

**DOI:** 10.3390/jcm11061526

**Published:** 2022-03-10

**Authors:** Min Seok Baek, Narae Lee, Jin Woo Kim, Jin Yong Hong

**Affiliations:** 1Department of Neurology, Wonju Severance Christian Hospital, Yonsei University Wonju College of Medicine, 20 Ilsan-ro, Wonju 26426, Korea; jinyhong@yonsei.ac.kr; 2Department of Nuclear Medicine, Wonju Severance Christian Hospital, Yonsei University Wonju College of Medicine, 20 Ilsan-ro, Wonju 26426, Korea; lnr84@yonsei.ac.kr; 3Department of Radiology, Wonju Severance Christian Hospital, Yonsei University Wonju College of Medicine, 20 Ilsan-ro, Wonju 26426, Korea; sunny-cocktail@hanmail.net

**Keywords:** Alzheimer’s disease, hippocampal subfields, amyloid-beta, biomarker

## Abstract

We investigated the relationship between hippocampal subfield volumes and cortical amyloid-beta (Aβ) deposition in Alzheimer’s disease (AD). Fifty participants (11 cognitively unimpaired [CU], 10 with mild cognitive impairment [MCI], and 29 with AD) who underwent ^18^F-florbetaben positron emission tomography, magnetic resonance imaging, and neuropsychological tests were enrolled. The hippocampal subfield volumes were obtained using an automated brain volumetry system with the Winterburn atlas and were compared among the diagnostic groups, and the correlations with the Aβ deposition and AD risk factors were determined. Patients with MCI and AD showed decreased volume in the stratum radiatum/lacunosum/moleculare (SRLM) of the cornu ammonis (CA)1 and CA4-dentate gyrus (DG) compared with the CU. Decreased SRLM and CA4-DG volumes were associated with an increased Aβ deposition in the global cortex (R = −0.459, *p* = 0.001; R = −0.393, *p* = 0.005, respectively). The SRLM and CA4-DG volumes aided in the distinction of AD from CU (areas under the receiver operating characteristic [AUROC] curve = 0.994 and 0.981, respectively, *p* < 0.001), and Aβ+ from Aβ− individuals (AUROC curve = 0.949 and 0.958, respectively, *p* < 0.001). Hippocampal subfield volumes demonstrated potential as imaging biomarkers in the diagnosis and detection of AD and Aβ deposition, respectively.

## 1. Introduction

The hippocampus plays a critical role in episodic and long-term memory through its extensive afferent and efferent connections with the cortical and subcortical structures [1,2]. Atrophy of the hippocampus is reportedly related to the clinical severity and progression of Alzheimer’s disease (AD), which typically presents as severe progressive memory impairment [3,4,5]. Therefore, hippocampal atrophy visualized using structural imaging has been used as one of the in vivo neuroimaging biomarkers in AD, together with the hallmark pathologic biomarkers, such as amyloid-beta (Aβ) and tau proteins [6,7].

The hippocampus consists of distinct anatomical regions called subfields—the cornu ammonis (CA) subfields 1–4, dentate gyrus (DG), and the subiculum. One of the main inputs to the hippocampus is the perforant pathway, in which information from the neocortex directly reaches the stratum radiatum/lacunosum/moleculare (SRLM), which includes the apical dendrites of CA1. The other indirect pathway conveys information from the neocortex to the DG. The information is subsequently conveyed via the mossy fibers to CA3, and it reaches the SRLM through Schaffer collaterals. The output from the hippocampus proceeds to the neocortex, including the entorhinal and prefrontal cortices, via diverse pathways [8]. The hippocampal subfields differentially support distinctive cognitive functions [9], and are affected by the risk factors of AD, such as age, sex, and the apolipoprotein-E (APOE) ε4 genotype [10]. Moreover, recent studies have demonstrated that patients with AD have distinctive patterns of atrophy in the hippocampal subfields [11,12]. Despite the massive interaction between the hippocampal subfields and cortical structures, the relationship between Aβ deposition in the neocortex and atrophy of the hippocampal subfields in AD is unclear.

While considering the patient selection in a study of the hippocampal subfields in AD, the pathological information on the presence of Aβ deposition is crucial. Since the diagnosis of AD or prodromal AD based on the clinical history and structural imaging is not always related to the pathologic findings [13], other types of dementia that also demonstrate hippocampal atrophy, such as limbic-predominant age-related TDP-43 encephalopathy [14] and Lewy body dementia [15,16], may result in unclear findings. Consequently, whether volumes of hippocampal subfields could aid in the determination of an accurate biomarker for the diagnosis of AD should be studied with participants with information of Aβ deposition.

In this study, we investigated the characteristics of hippocampal subfields volumes among patients who have information of Aβ deposition. Using an automated web-based volumetry system, segmentation of the hippocampal subfields was performed in a more elaborate manner compared to that in previous studies, in which, the SRLM of CA1 with abundant synapses from the neocortex was separately segmented from the stratum pyramidale of CA1. Moreover, we also tested the hippocampal subfield volumes as biomarkers for AD diagnosis and the presence of Aβ deposition.

## 2. Materials and Methods

### 2.1. Participants

We retrospectively included 11 cognitively unimpaired (CU) individuals, 10 patients with mild cognitive impairment (MCI), and 29 patients with AD at the Memory Disorder Clinic of Wonju Severance Christian Hospital between January 2021 and November 2021. All the participants underwent brain magnetic resonance imaging (MRI), ^18^F-florbetaben positron emission tomography (PET), neuropsychiatric tests, and APOE genotyping. Diagnostic criteria for probable AD dementia with evidence of the AD pathophysiological process was implemented based on the National Institute of Neurological and Communicative Disorders and Stroke [17]. Patients with MCI were defined by the National Institute on Aging—Alzheimer’s Association criteria for MCI due to AD with high likelihood [18]. CU was defined as those who showed normal performance on baseline neuropsychological tests, no abnormalities on brain MRI, and negative results of cerebral Aβ accumulation in the ^18^F-florbetaben PET scan. Aβ-positivity was determined by a nuclear medicine specialist using the validated visual assessment methods [19,20].

### 2.2. Acquisition of PET and MR Images

Using a Discovery STe PET/computed tomography (CT) scanner (GE Healthcare, Milwaukee, WI, USA), PET images were acquired for 15 min at 90 min following the ^18^F-florbetaben injection. CT images were acquired for attenuation correction prior to the PET scan. Finally, three-dimensional PET images were reconstructed in a 256 × 256 × 223 matrix with 2.34 × 2.34 × 3.2 mm^3^ voxel size using the ordered-subsets expectation maximization algorithm. Sagittal T1-weighted brain MR images were acquired by magnetization prepared rapid acquisition gradient-echo (MPRAGE) volumetric scans (3T Siemens MRI, Siemens, Erlangen, Germany) with voxel spacing of 0.4 × 0.4 × 0.9 mm^3^.

### 2.3. Image Processing Steps

Hippocampal subfield segmentation was performed using the hippocampus subfield segmentation (HIPS) pipeline, based on a label fusion segmentation method and neural-network-based error correction [21]. This pipeline is publicly available online (http://volbrain.upv.es, accessed on 7 March 2022). In this study, we performed the analysis using T1 sagittal images with the Winterburn atlas [22], which produces five hippocampal subfield labels based on the morphology and intensity of densely myelinated molecular layers: CA1, CA2–3, CA4-DG, SRLM, and subiculum (Figure 1).

For quantifying the cerebral Aβ deposition, T1-weighted MR images were processed using the FreeSurfer 6.0 (Massachusetts General Hospital, Boston, MA, USA) software for creating participant-specific volume-of-interest (VOIs) mask images, as described in a previous study [23]. Cortical regions were parcellated using curvature information, and subcortical regions were segmented with the probabilistic registration method. By merging anatomically related regions, participant-specific composite VOI images for cortical and subcortical regions were created. The voxel count for each region was considered as the regional cortical volume. Statistical parametric mapping 12 (Wellcome Trust Centre for Neuroimaging, London, UK) and an in-house software implemented in MATLAB R2015b (MathWorks, Natick, MA, USA) were used to process the PET images and measure the regional uptake values. PET images were co-registered to individual MR images within the FreeSurfer space, and standardized uptake value ratio (SUVR) images were then created with the cerebellar crus median obtained by overlaying the template cerebellar crus VOI on the spatially normalized PET images. Finally, by overlaying the participant-specific composite VOI, regional SUVR values were measured.

### 2.4. Statistical Analysis

SPSS 26 (IBM Corp., Armonk, NY, USA) was used for the statistical analysis. For group comparisons based on the demographic characteristics, two-sample *t*-tests and chi-square tests were used for the continuous and categorical demographic data, respectively. The hippocampal subfield volumes were compared between the diagnostic groups using analysis of variance (ANOVA), and the group differences were further analyzed after adjusting for age, sex, presence of APOE ε4, and intracranial volume (ICV) using the general linear model. Pearson’s correlation method was used for the correlation analysis between the hippocampal subfields volumes and regional cortical uptake of ^18^F-florbetaben. Region-wise multiple comparisons were corrected using the Benjamini–Hochberg’s false discovery rate method [24]. Multivariable linear regression model was used to analyze the effects of the risk factors on the hippocampal subfield volumes. Sensitivity and specificity were generated with the receiver operating characteristic (ROC) curve analysis using the method of DeLong et al. [25], implemented in Medcalc 17.2 (MedCalc Software, Ostend, Belgium). Youden’s method was used to identify an optimal cut-off point of the ROC curves.

## 3. Results

### 3.1. Demographic Characteristics

Detailed demographic characteristics of the study participants are summarized in Table 1. The patients with MCI and AD were older than the CU individuals. APOE ε4 carriers were more frequent in patients with AD and MCI than in CU individuals. The total hippocampal volume demonstrated decreasing patterns with increased clinical severity. Distribution by sex and the duration of education were not different among the diagnostic groups. SUVR for ^18^F-florbetaben in the global cortex was greater in patients with AD and MCI than in the CU individuals.

### 3.2. Hippocampal Subfields Volumes

The total hippocampus volume and the volumes of all the hippocampal subfields except the subiculum were different among the diagnostic groups (Figure 2). The group comparisons between Aβ+ and Aβ− participants also showed differences in the volumes of all hippocampal subfields except the subiculum (Appendix A). There were differences in the volumes of the hippocampus, CA4-DG, and SRLM between CU and MCI and between CU and AD (Figure 2A,D,E). Significant differences were noted in the volumes of CA2-3 and CA4 between MCI and AD and between CU and AD (Figure 2B,C). The mean volume of the subiculum was decreased in patients with AD, followed by patients with MCI and CU with no statistical significance (Figure 2F).

After adjustment for age, sex, presence of the APOE ε4 genotype, and ICV, the volumes of the hippocampus and CA1 were greater in the CU individuals than in patients with AD (Appendix A). The CA4-DG volume was greater in the CU individuals (1.26 ± 0.16 cm^3^) than in patients with MCI (0.87 ± 0.28 cm^3^) and AD (0.70 ± 0.25 cm^3^) after adjustment for covariates. The SRLM volume was also greater in the CU individuals (0.91 ± 0.13 cm^3^) than in patients with MCI (0.58 ± 0.27 cm^3^) and AD (0.42 ± 0.18 cm^3^) following the adjustment for the covariates. The mean volume of CA2–3 and the subiculum was decreased in patients with AD, followed by patients with MCI and the CU individuals. However, the results were not significant after adjusting for covariates (Appendix A).

### 3.3. Correlation between Hippocampal Subfield Volumes and Regional Aβ Deposition

The decreased CA1 volume demonstrated a non-significant association with the increased Aβ deposition in the inferior parietal (R = −0.253 *p* = 0.077) and inferior temporal cortices (R = −0.246 *p* = 0.085) (Table 2). The decreased CA2–3 volume showed a non-significant association with the increased Aβ deposition in the inferior parietal cortex (R = −0.269 *p* = 0.059). However, the decreased CA4-DG and SRLM volumes were associated with increased Aβ deposition in the global, prefrontal, sensorimotor, parietal, precuneus, occipital, temporal, cingulate, and insular cortices. All of the regions survived multiple comparisons. The subiculum volume did not reveal an association with regional Aβ deposition (Table 2).

### 3.4. ROC Curves of the Hippocampal Subfield Volumes

Upon comparison between CU and AD, the SRLM volume revealed the highest area under the curve (AUC) value among the hippocampal subfields. The CA4-DG and CA1 volumes showed an AUC value higher than 0.9, and the CA2-3 volume demonstrated an AUC value higher than 0.8 (Figure 3A). The hippocampal subfield volumes demonstrated a good performance in distinguishing between Aβ+ and Aβ− individuals. The SRLM and CA4-DG volumes demonstrated an AUC value higher than 0.9, and the CA1 volume demonstrated an AUC value higher than 0.8 (Figure 3B). The SRLM and CA4-DG volumes demonstrated a higher AUC value than the whole hippocampus volume in the diagnosis of AD, and in the detection of Aβ positivity (Figure 3A,B).

### 3.5. Multiple Regressions Analysis

Age was associated with decreased volumes in the CA1, CA4-DG, and SRLM. Lower Mini-Mental State Examination scores were associated with decreased CA1, CA2–3, CA4-DG, and SRLM volumes (Table 2). Notably, the presence of the APOE4 allele was associated with a decreased volume in the CA1 (β = −0.193, *p* = 0.033). The duration of education and sex were not associated with the hippocampal subfield volumes (Table 3).

## 4. Discussion

In this study, we demonstrated that the automatically measured volumes of SRLM and CA4-DG in the hippocampal subfields were associated with Aβ deposition in the diffuse cortical regions.

CA1 in the hippocampus is more likely to be affected in AD, as demonstrated in a histopathological study [26]. Pathologic studies have shown that SRLM among the hippocampal subfields is one of the earliest sites involved in AD. In an AD transgenic mouse model study, synaptic loss within the SRLM preceded both neuronal death and the appearance of histopathologic changes [27]. In humans, neuroimaging studies using 7T MR scans revealed that the atrophy in the SRLM of CA1 is more greatly affected than the stratum pyramidale of CA1 in AD [28,29]. Some studies demonstrated that patients with MCI also revealed a volume reduction in the CA1 region [30,31]. However, the effect of the atrophy in the SRLM was not evaluated since the studies defined the pyramidal layer and SRLM of CA1 as a whole unit. Our results suggest that CA1 atrophy in the MCI is also mainly impacted by SRLM atrophy in CA1. Therefore, hippocampal subfield volumes could be beneficial in the early diagnosis of AD.

The relationship between the hippocampal subfield volumes and cortical Aβ deposition was indirectly assessable from the group comparison results in previous studies. In CU individuals, Aβ+ individuals showed reduced volumes in the hippocampal tail and subiculum and a difference in the CA1 volume was not observed [32]. The mean value of the CA1 volume was decreased in Aβ+ MCI compared to Aβ− MCI in the study. However, the results did not show statistical significance [31]. The findings could be attributed to the small sample size, and, more importantly, both studies segmented the stratum pyramidale and SRLM of CA1 as a whole unit. Our study suggests that the SRLM and CA4-DG are specific hippocampal subfields that are associated with cortical Aβ deposition. The SRLM of CA1 and CA4-DG are the two main regions that receive the input from the neocortex via the perforant pathway [8]. From the neocortex, the hippocampus transmits two inputs. One is the projection to the DG and the other is the projection to the SRLM of the CA1, both conveyed by the medial and lateral entorhinal cortex [33]. Alteration of the neocortex attributed to Aβ deposition may induce neurodegeneration of the first post-synaptic recipient region through the disruption of the corticohippocampal connections in AD.

Interestingly, APOE-ε4 positivity showed an association only with the CA1 volumes in the hippocampus, rather than other regions showing a higher diagnostic performance. The cell bodies of the stratum pyramidale are more likely to be affected by genetic factors, whereas SRLM and CA4-DG are relatively more affected through the neural connections from the neocortex. In a transgenic mice study, the APOE-ε4 genotype was related to the decreased size of the pyramidal cells in CA1 [34]. The shortened dendritic length and decreased neuronal density of the CA1 neurons in APOE-ε4 transgenic mice were mainly located in the proximal segments of the apical dendrite of CA1 [35]. Atrophy of the pyramidal cell bodies of CA1 and the proximal segment of apical dendrites may affect the decreased CA1 volume in the APOE-ε4 carriers in this study.

Our study has several limitations. First, the mean age difference between CU and AD (11.3 years) may affect the results in this study. Although we showed the group comparison results with adjustment for covariates, age itself may have associations with other risk factors of AD (i.e., vascular risk factors), which were not analyzed in the study. Second, volume measurement using MRI has a possible inherent bias of over- or underestimation in term of precise delineation and quantification. The ranges of the hippocampal subfield volumes acquired by the Winterburn atlas in this study might be different from previous data obtained by manual segmentation, owing to labeling protocol differences [36]. Third, the relatively small number of patients could affect the power of the study to detect all of the differences between the diagnostic groups. Fourth, the cross sectional design of the study makes it impossible to draw conclusions on the effect of time on the changes in hippocampal atrophy affected by the risk factors of AD, and the changes in cognitive function affected by the changes in the hippocampal subfield volumes. Lastly, the benefits of the hippocampal subfield volumes in AD diagnosis and Aβ positivity detection could be underestimated, as the total volume of the hippocampus also shows a high performance. However, the automated segmentation protocol quantifies the volumes of the hippocampal subfields and the total hippocampus in a single process with a short processing time. Therefore, the use of the hippocampal subfields volumes as neuroimaging biomarkers for AD results in a greater accuracy compared to the visual scale method, and is also more convenient than the manual segmentation protocol.

## 5. Conclusions

Distinct hippocampal subfield volumes are affected by the pathologic changes and risk factors of AD. Hippocampal subfield volumes may be useful in AD diagnosis and Aβ positivity detection.

## Figures and Tables

**Figure 1 jcm-11-01526-f001:**
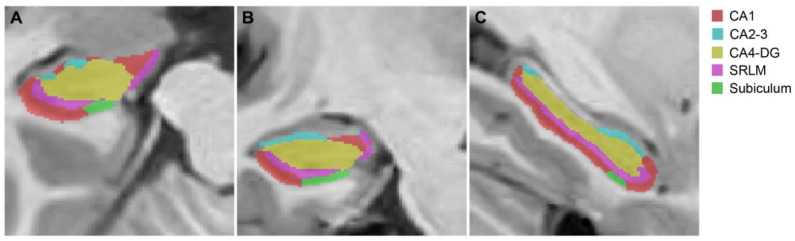
An example of hippocampal subfield segmentation in the HIPS pipeline. (**A**) Axial, (**B**) Coronal, and (**C**) Sagittal sections. Abbreviations: CA = cornu ammonis, DG = dentate gyrus, SRLM = stratum radiatum/lacunosum/moleculare.

**Figure 2 jcm-11-01526-f002:**
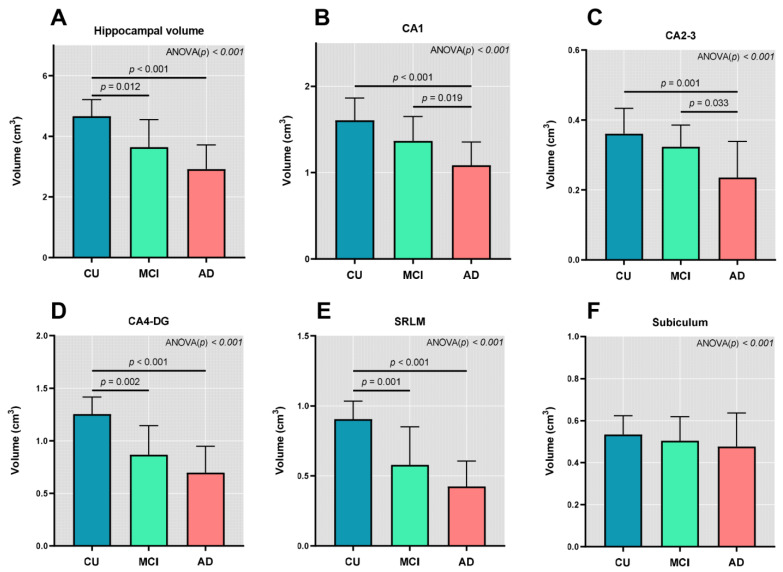
Between-group comparisons of the hippocampal subfield volumes. (**A**) Total hippocampal volume, (**B**) CA1, (**C**) CA2–3, (**D**) CA4-DG, (**E**) SRLM, (**F**) subiculum. Abbreviations: AD = Alzheimer’s disease, CU = cognitively unimpaired, MCI = mild cognitive impairment, CA = cornu ammonis, SRLM = stratum radiatum/lacunosum/moleculare.

**Figure 3 jcm-11-01526-f003:**
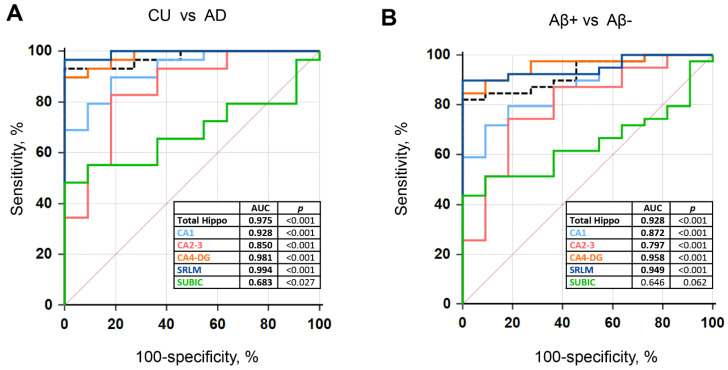
Receiver operating characteristic (ROC) curves of the hippocampal subfield volumes. (**A**) AD from CU (**B**) Aβ+ vs. Aβ−. Abbreviations: CU = cognitively unimpaired, AD = Alzheimer’s disease, Aβ = Amyloid-β, Total Hippo = whole hippocampus volume.

**Table 1 jcm-11-01526-t001:** Baseline demographic characteristics.

	CU	MCI	AD
N	11	10	29
Age (years)	65.8 ± 7.3	73.8 ± 9.4 ^a^	77.1 ± 8.4 ^b^
Sex (M:F)	6:5	7:3	13:16
Duration of education (years)	9.9 ± 5.0	10.6 ± 4.0	10.1 ± 5.1
APOE ε4 genotype	0/11	3/10 ^a^	12/29 ^b^
MMSE	27.5 ± 1.6	24.7 ± 2.5 ^a^	20.6 ± 4.6 ^b,c^
CDR-SB	0	2.0 ± 0.7 ^a^	4.6 ± 2.6 ^b,c^
Intracranial Volume (cm^3^)	1361 ± 134	1482 ± 140	1313 ± 106 ^c^
Volume of hippocampus (cm^3^)	4.66 ± 0.55	3.64 ± 0.91 ^a^	2.92 ± 0.80 ^b,c^
^18^FBB SUVR for global cortex	1.39 ± 0.10	1.91 ± 0.45 ^a^	1.86 ± 0.35 ^b^

Data are presented as mean ± SD. ^a^ *p* < 0.05 for the comparisons between the CU and MCI. ^b^ *p* < 0.05 for the comparisons between the CU and AD. ^c^ *p* < 0.05 for the comparisons between MCI and AD. Abbreviations: CU = cognitively unimpaired, MCI = mild cognitive impairment, AD = Alzheimer’s disease, APOE = apolipoprotein-E, MMSE = Mini-Mental State Examination, CDR-SB = Clinical Dementia Rating—Sum of Boxes, ^18^FBB = ^18^F-florbetaben, SUVR = standardized uptake value ratio.

**Table 2 jcm-11-01526-t002:** Correlation of the hippocampal subfield volumes with the regional ^18^F-florbetaben standardized uptake value ratio in all of the participants.

	CA1		CA2–3		CA4-DG		SRLM		SUBICULUM	
	R	*p*	R	*p*	R	*p*	R	*p*	R	*p*
Global cortex	−0.186	0.195	−0.218	0.129	**−0.393**	0.005	**−0.459**	0.001	0.012	0.934
Prefrontal	−0.173	0.228	−0.238	0.097	**−0.405**	0.004	**−0.455**	0.001	0.006	0.967
Sensorimotor	−0.120	0.408	−0.207	0.150	**−0.333**	0.018	**−0.379**	0.007	0.054	0.710
Superior parietal	−0.197	0.171	−0.235	0.101	**−0.392**	0.005	**−0.446**	0.001	0.008	0.957
Inferior parietal	−0.253	0.077	−0.269	0.059	**−0.435**	0.002	**−0.499**	<0.001	−0.035	0.812
Precuneus	−0.203	0.157	−0.232	0.105	**−0.409**	0.003	**−0.480**	<0.001	0.030	0.836
Occipital	−0.216	0.132	−0.119	0.409	**−0.337**	0.017	**−0.379**	0.007	−0.085	0.558
Superior temporal	−0.181	0.207	−0.200	0.164	**−0.354**	0.012	**−0.456**	0.001	0.056	0.701
Middle temporal	−0.216	0.131	−0.235	0.100	**−0.409**	0.003	**−0.485**	<0.001	0.050	0.731
Inferior temporal	−0.246	0.085	−0.208	0.147	**−0.402**	0.004	**−0.495**	<0.001	0.011	0.940
Entorhinal	0.174	0.226	0.171	0.235	−0.012	0.932	−0.124	0.390	0.215	0.134
Parahippocampal	0.058	0.688	−0.007	0.962	−0.164	0.254	−0.274	0.054	0.186	0.197
Amygdala	0.103	0.475	0.175	0.223	−0.011	0.938	−0.130	0.369	0.028	0.846
Anterior cingulate	−0.103	0.475	−0.168	0.243	**−0.329**	0.020	**−0.413**	0.003	0.054	0.708
Posterior cingulate	−0.149	0.303	−0.158	0.274	**−0.368**	0.009	**−0.451**	0.001	0.029	0.839
Insula	−0.127	0.381	−0.172	0.232	**−0.326**	0.021	**−0.411**	0.003	0.037	0.797

Bold numbers represent the regions that survived correction for region-wise multiple comparisons (false discovery rate-corrected *p* < 0.05). Abbreviations: CA = cornu ammonis, DG = dentate gyrus, SRLM = stratum radiatum/lacunosum/moleculare.

**Table 3 jcm-11-01526-t003:** Effects of the risks of Alzheimer’s disease on the hippocampal subfield volumes.

	CA1		CA2-3		CA4-DG		SRLM		SUBICULUM	
	B (SE)	*p*	B (SE)	*p*	B (SE)	*p*	B (SE)	*p*	B (SE)	*p*
Age	**−0.011 (0.005)**	0.036	−0.004 (0.002)	0.050	**−0.013 (0.005)**	0.010	**−0.011 (0.004)**	0.007	−0.002 (0.003)	0.594
Female	−0.090 (0.089)	0.320	−0.012 (0.033)	0.710	0.002 (0.083)	0.983	−0.021 (0.069)	0.761	−0.087 (0.052)	0.107
APOE4	**−0.193 (0.086)**	0.033	−0.009 (0.032)	0.786	−0.034 (0.080)	0.675	−0.049 (0.066)	0.468	−0.080 (0.051)	0.125
Education	0.004 (0.010)	0.700	−0.004 (0.004)	0.266	−0.008 (0.009)	0.383	−0.003 (0.008)	0.716	0.003 (0.006)	0.564
MMSE	**0.030 (0.010)**	0.005	**0.011 (0.004)**	0.005	**0.032 (0.009)**	0.002	**0.023 (0.007)**	0.004	0.010 (0.006)	0.088

Bold number indicates risk factors with statistical significance (*p* < 0.05). Abbreviations: APOE = apolipoprotein-E, MMSE = Mini-Mental State Examination, CA = cornu ammonis, DG = dentate gyrus, SRLM = stratum radiatum/lacunosum/moleculare, SE = standard error.

## Data Availability

Data generated by this study are available from the corresponding author on reasonable request. The data are not publicly available due to privacy restriction.

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
