# Peer review of "Association of Hippocampal Subfield Volumes with Amyloid-Beta Deposition in Alzheimer’s Disease"

_jcm, 2022, doi:10.3390/jcm11061526_

Round 1

Reviewer 1 Report

The authors present an interesting study about hyppocampal subfield volumes in patients with AD. 

The main problem is the big age difference between control subjects and people with AD. Even if they adjust every measurement by age, a 12 year mean difference is too high. It should be acknowledge in the limitations of the study. 

The clinical implications of the study are somewhat blurry. After reading the discussion I am not convinced that a subfield hippocampal analysis is better or faster to do than a whole hippocampal volumetry or visual analysis. I recommend to downplay the clinical implications an focus in the  pathophysiologocal relevance.

Author Response

Reviewer #1

1. The main problem is the big age difference between control subjects and people with AD. Even if they adjust every measurement by age, a 12 year mean difference is too high. It should be acknowledged in the limitations of the study.

Our Answer: We completely agree with the reviewer’s comments. Although we adjusted age as one of covariates in the analysis, it still has associations with other risk factors of AD and age related comorbidities that can affect the hippocampal atrophy. It would be solved by large-sized controlled study; however, we admit that this is one of the limitations to our study. We have added the following sentences in the discussion section.

In the Discussion section (line 257)

→ Our study has several limitations. First, the mean age difference between CU and AD (11.3 years) could affect the results in this study. Although we presented the group comparison results while adjusting for the covariates, age itself may have associations with other risk factors of AD (i.e., vascular risk factors), which were not analyzed in the study.

2. The clinical implications of the study are somewhat blurry. After reading the discussion I am not convinced that a subfield hippocampal analysis is better or faster to do than a whole hippocampal volumetry or visual analysis. I recommend to downplay the clinical implications an focus in the pathophysiologocal relevance.

Our Answer: Thank you for the comment. We realized that some sentences in the discussion section were redundant and ambiguous, which could confuse the readers as noted by the reviewer. We omitted and tone-downed certain sentences as follows.

In the Discussion section (line 206)

→ Omitted: The SRLM and CA4-DG volumes showed greater utility in the diagnosis of AD and the detection of Aβ positivity.

In the Discussion section (line 235)

→ Omitted: Consequently, the information of the hippocampal subfields could be a neuroimaging biomarker for the diagnosis of AD, as well as for the detection of Aβ positivity.

In the Discussion section (line 270)

→ Lastly, the benefits of the hippocampal subfield volumes in AD diagnosis and Aβ positivity detection could be underestimated, as the total volume of the hippocampus also shows high performance. However, the automated segmentation protocol quantifies the volumes of the hippocampal subfields and the total hippocampus in a single process with short processing time. Therefore, use of the hippocampal subfields volumes as neuroimaging biomarkers for AD results in a greater accuracy compared to the visual scale method and is also more convenient than the manual segmentation protocol.

In the Conclusions section (line 278)

→ Hippocampal subfield volumes may be useful in AD diagnosis and Aβ positivity detection.

Reviewer 2 Report

This article aimed to test the feasibility of hippocampal subfield volumes as biomarkers for AD diagnosis and intracranial Aβ deposition. It is an innovation point that separating SRLM from CA1 as an independent segmentation. The patient enrollment criteria is rigorous. When doing correlation analysis, the cortical division is very detailed. However, this paper needs some improvement before acceptance for publication. My detailed comments are as follows:

  1. General linear model was used to adjust the variables and to compare the hippocampal subfield volumes between CU, MCI and AD group, which may reduce the persuasiveness of the outcome. A larger sample size is needed and the age, sex and other comorbidities which related to the encephalatrophy should be balanced at baseline. Variables like volume of hippocampus, APOE ε4 genotype, intracranial volume need not to be balanced since they relate to the diagnosis itself. Then ANOVA could be used to compare the deference between groups with a higher persuasiveness.
  2. The ROC curve of the whole hippocampus volume should be drawn. From the perspective of convenience, is it more convenient to detect the whole hippocampus volume rather than the subfield volumes if the AUC value of hippocampus volume curve is at a high level?
  3. The data comparison between the Aβ+ and Aβ- group could be described and regression could be used to analyze the risk factors for Aβ+.
  4. Little mistake like “HC” in the annotation of table 1.

Author Response

Response to reviewer

Reviewer #2

1. General linear model was used to adjust the variables and to compare the hippocampal subfield volumes between CU, MCI and AD group, which may reduce the persuasiveness of the outcome. A larger sample size is needed and the age, sex and other comorbidities which related to the encephalatrophy should be balanced at baseline. Variables like volume of hippocampus, APOE ε4 genotype, intracranial volume need not to be balanced since they relate to the diagnosis itself. Then ANOVA could be used to compare the deference between groups with a higher persuasiveness.

Our Answer: We completely agree with the reviewer’s comments. We revised the group comparisons (Figure 2) with ANOVA (P) and p-value for simple mean comparisons without covariates to improve the impact. The results of the group comparisons with covariates have been included in the Supplement materials as a table.

In the Results section (line 150)

→ The total hippocampus volume and the volumes of all the hippocampal subfields except the subiculum were different among the diagnostic groups (Figure 2). The group comparisons between Aβ+ and Aβ- participants also showed differences in the volumes of all the hippocampal subfields except the subiculum (Supplement Figure S1). There were differences in the volumes of the hippocampus, CA4-DG, and SRLM between CU and MCI and between CU and AD (Figure 2ADE). Differences were noted in the volumes of CA2-3 and CA4 between MCI and AD and between CU and AD (Figure 2BC). The mean volume of the subiculum was decreased in patients with AD followed by patients with MCI and CU with no statistical significance (Figure 2F).

Supplemental Table S1. Hippocampal subfield volumes in the diagnostic groups

In the Materials and Methods section (line 122)

Using the general linear model, the hippocampal subfield volumes were compared between the diagnostic groups after adjusting for age, sex, presence of APOE ε4, and intracranial volume (ICV).

→ The hippocampal subfield volumes were compared between the diagnostic groups using analysis of variance (ANOVA), and the group differences were further analyzed after adjusting for age, sex, presence of APOE ε4, and intracranial volume (ICV) using the general linear model.

 2. The ROC curve of the whole hippocampus volume should be drawn. From the perspective of convenience, is it more convenient to detect the whole hippocampus volume rather than the subfield volumes if the AUC value of hippocampus volume curve is at a high level?

Our Answer: Thank you for this valuable comment.

First of all, we sincerely apologize for the unintended error in Figure 3. Upon re-analyzing as per the reviewer’s comment, we realized that the data used for ROC analysis in Figure 3 include Aβ-positive CU (n=2), as well as Aβ-negative MCI (n=16), and AD (n=3), who did not meet the participants’ inclusion criteria in this study. Although these results indicate that hippocampal subfield volumes can be applied in the clinical setting for patients without information of the Aβ burden, we consider that the ROC analysis results only with the participants who meet the inclusion criteria (Aβ-negative CU, and Aβ-positive MCI and AD) can convey a clear message in this study. Therefore, we have revised figure 3, and also included the information of the entire hippocampus volume.

As the reviewer has mentioned, the entire hippocampus volume also showed high value for discriminating between CU vs. AD (AUC 0.975) or Aβ+ vs. Aβ- individuals (AUC 0.928). We partly agree with the reviewer that the total hippocampal volume is also a good biomarker for AD as previously reported. However, despite the small volume fraction of SRLM (approximately 15% of the total hippocampus volume) and CA4-DG (approximately 25% of the total hippocampus volume) compared with the entire hippocampus volume, we consider that these two regions provide more accurate diagnostic information for AD.

We presume the web-based volumetry system, which requires less time consumption than the conventional freesurfer system, to be a convenient method. As the total hippocampus volume and subfield volumes are assessable in one single process, we recommend SRLM and CA4-DG volumes as more accurate biomarkers for AD. We have revised the ROC analysis results. We have also toned down the clinical implications in the Discussion section as per the recommendations by Reviewer 1 (comment#2).

In the Results section (line 191)

→ Upon comparison between CU and AD, the SRLM volume revealed the highest area under the curve (AUC) value among the hippocampal subfields. The CA4-DG and CA1 volumes showed an AUC value higher than 0.9, and CA2-3 volume demonstrated an AUC value higher than 0.8 (Figure 3A). The hippocampal subfield volumes showed good performance in distinguishing Aβ+ individuals from Aβ- individuals. The SRLM and CA4-DG volumes demonstrated an AUC value higher than 0.9, and CA1 volume demonstrated an AUC value higher than 0.8 (Figure 3B). The SRLM and CA4-DG volumes demonstrated a higher AUC value than the whole hippocampus volume in the diagnosis of AD and in the detection of Aβ positivity (Figure 3AB).

Figure 3 (Revised). Receiver operating characteristic (ROC) curves of the hippocampal subfield volumes. (A) AD from CU (B) Aβ+ vs. Aβ-. Abbreviations: CU = cognitively unimpaired, AD = Alzheimer’s disease, Aβ = Amyloid-β, Total Hippo = whole hippocampus volume

In the Discussion section (line 270)

→ Lastly, the benefits of the hippocampal subfield volumes in AD diagnosis and Aβ positivity detection could be underestimated, as the total volume of hippocampus also shows high performance. However, the automated segmentation protocol quantifies the volumes of the hippocampal subfields and the total hippocampus in a single process with short processing time. Therefore, use of the hippocampal subfields volumes as neuroimaging biomarkers for AD results in a greater accuracy compared to the visual scale method and is also more convenient than the manual segmentation protocol.

3. The data comparison between the Aβ+ and Aβ- group could be described and regression could be used to analyze the risk factors for Aβ+.

Our Answer: Thank you for this valuable comment. We added the results of the group comparisons between Aβ+ and Aβ- in the supplementary materials to enhance the readers’ understanding. Unfortunately, to analyze the risk factors for Aβ positivity, larger sample size, especially more information on the Aβ- MCI and AD, as well as Aβ+ CU, who were not included in this study, is required to void selection bias. We added the information on the group comparisons between Aβ+ and Aβ- in the manuscript as follows.

In the Results section (line 151)

→ The group comparisons between Aβ+ and Aβ- participants also showed differences in the volumes of all hippocampal subfields except the subiculum (Supplement Figure S1).

Supplement Figure S1. The hippocampal subfield volumes in Aβ+ and Aβ- participants. 

4. Little mistake like “HC” in the annotation of table 1.

Our Answer: We have revised it. Thank you for the valuable comment.
